# Valorization of Citrus Reticulata Peels for Flavonoids and Antioxidant Enhancement by Solid-State Fermentation Using *Aspergillus niger* CGMCC 3.6189

**DOI:** 10.3390/molecules27248949

**Published:** 2022-12-15

**Authors:** Daniel Mamy, Yuanyuan Huang, Nelson Dzidzorgbe Kwaku Akpabli-Tsigbe, Maurizio Battino, Xiumin Chen

**Affiliations:** 1School of Food and Biological Engineering, Jiangsu University, 301 Xuefu Road, Jingkou District, Zhenjiang 212013, China; 2Institute of Food Physical Processing, Jiangsu University, 301 Xuefu Road, Jingkou District, Zhenjiang 212013, China; 3International Joint Research Laboratory of Intelligent Agriculture and Agri-Products Processing, Jiangsu University, Zhenjiang 212013, China; 4Higher Institute of Sciences and Veterinary Medicine (ISSMV) of Dalaba, Dalaba-Tangama P.O. Box 09, Guinea; 5Department of Nutrition and Food Science, College of Basic and Applied Sciences, University of Ghana, Legon P.O. Box LG 134, Ghana; 6Department of Clinical Sciences, Università Politecnica delle Marche, 60100 Ancona, Italy

**Keywords:** citrus peels, solid-state fermentation, polymethoxylflavones, nobiletin, tangeretin, antioxidant properties

## Abstract

The bioactive components and bioactivities of citrus peel can be enhanced with microbial fermentation. Accordingly, this study investigated the ability of *Aspergillus niger* CGMCC3.6189 to accumulate flavonoids in Citrus reticulata peel powder (CRPP) by solid-state fermentation (SSF). Under the optimal SSF conditions including 80% moisture, 30 °C, pH 4.0, 4 × 10^7^ spores/g d.w. CRPP, and 192 h, the total phenolic content (TPC), total flavonoid content (TFC), and 2,2′-azinobis-(3-ethylbenzothiazoline-6-sulfonic acid) (ABTS) and 1,1-diphenyl-2-picrylhydrazyl radical (DPPH) scavenging activities of fermented CRPP significantly increased by 70.0, 26.8, 64.9, and 71.6%, respectively. HPLC analysis showed that after fermentation, the contents of hesperidin, nobiletin, and tangeretin were significantly increased from 19.36, 6.31, and 2.91 mg/g to 28.23, 7.78, and 3.49 mg/g, respectively, while the contents of ferulic acid and narirutin were decreased under the optimal fermentation conditions. Fermentation time is one of the most important factors that affect the accumulation of flavonoids and antioxidant activity; however, extended fermentation time increased the darkness of CRPP color. Therefore, our study provides a feasible and effective SSF method to increase the bioactive components and the antioxidant activity of CRPP that may be used in food, nutraceutical, and medicinal industries.

## 1. Introduction

Citrus is a group of the most abundant fruits in terms of production and growing area. China is the largest tangerine (*Citrus reticulate*) producer worldwide, with an annual production of 23.12 million tonnes (MT) in 2020, accounting for about 60% of global production [1]. This production results in a significant accumulation of waste, which creates a major management challenge in developing countries [2]. 

The dried peel of the citrus fruit (Pericarpium *citri reticulatae*, PCR) known as Chenpi exhibits various beneficial functional properties such as anti-inflammatory, anti-diabetic, anticancer, anti-microbial, anti-viral, antioxidative, antimutagenic, and anti-glycemic activities due to their richness in polyphenols, carotenoids, vitamins, and fiber [3,4]. Over eighty flavonoids classified into flavones, flavonols, flavanones, polymethoxylflavones, and anthocyanins are found in citrus fruits, among which naringin, hesperidin, narirutin, neo-hesperidin, nobiletin, tangeritin, hesperetin, and naringenin are most abundant in citrus peels [5,6]. It is generally known that “the longer time Chenpi is stored, the better health benefits it has”. Aging increases the accumulation of phenolics and flavonoids in Chenpi [7], therefore promoting the health benefits of aged citrus peels. However, it is not necessarily true that the longer the storage time, the higher the flavonoid compositions. The PCR metabolite levels increased within 3–15 years of storage, while showing a decrease trend to a stable state after storing for 15–30 years [8]. Yang et al. [9] found that the antioxidant activity of Chenpi reached a maximum in the 5-year-old Chenpi and then decreased with the extended storage time. Therefore, storage time is critical for preparing high-quality Chenpi.

The natural aging process of Chenpi generally takes place in moisture-controlled conditions for many years, during which flavonoids accumulate due to microbial biotransformation. The genera of microbes identified in Chenpi include *Penicillium citrinum*, *Penicillium milmonillium*, *Penicillium common*, *Aspergillus flavus*, *Aspergillus niger*, *Penicillium minioluteum* [7], *Bacillus, Lactococcus, Pseudomonas, Oceanobacillus, Pseudarthrobacter, Enterococcus*, and *Psychrobacter* [10], among which *Bacillus* and *Lactococcus* are the two main genera [10]. It is believed that the microbes significantly improve the chemical quality of Chenpi. Currently, microbial processes are being developed to biotransform steroids and flavonoids for direct use or as precursors for new drugs and other beneficial compounds [11]. Various microbes have been used to accelerate the biotransformation of flavonoids, among which *A. niger* is one of the most used microorganisms [12]. For example, flavone was hydroxylated to 4’-hydroxyflavone and subsequently to 3’,4’-dihydroxyflavone by *A. niger* ATCC 43949. *A. niger* NRRL 2295 and *A. niger* X172 also hydroxylated flavone to 4’-hydroxyflavone [11].

The inoculation of *A. niger* isolated from Citrus reticulata peel (Chenpi) using solid-state fermentation (SSF) increased the total flavonoid content (TFC) and the flavonoid aglycones such as hesperetin and naringenin, while the corresponding flavanone glycosides (hesperidin and narirutin) were decreased in a much shorter period compared with the natural aging process [7]. SSF is an effective, environmentally friendly, cost-effective, and feasible approach that has been used to increase the concentration of bioactive compounds and antioxidant activity in agro-industrial wastes and plant by-products [13]. It is an effective technique to increase the concentration of phenols and flavonoids [14]. Consequently, it could be an asset for the accumulation of flavonoids in *Citrus reticulata* peel by *A. niger* strains. The biotransformation of flavonoids using *A. niger* has been widely reported [12]; however, studies associated with the changes in the phytochemical profile, color, and antioxidant activity of the citrus peels under the SSF by *A. niger* are scarce. Accordingly, this study aimed to evaluate the potential of increasing flavonoid compounds and antioxidant activity in *Citrus reticulata* peel by *A. niger* CGMCC 3.6189 under different SSF conditions including pH, temperature, moisture content, inoculation concentration, and fermentation time. In addition, the changes in the color of citrus peel were also assessed. Our study provides valuable information for the valorization of citrus peel waste.

## 2. Results

### 2.1. Effect of pH on TPC, TFC, Antioxidant Activity, and Phytochemical Compositions in CRPP

Table 1 shows the effects of the inoculum pH on the TPC, TFC, and ABTS and DPPH scavenging capacities of CRPP before and after SSF. The TPC, TFC, and ABTS and DPPH of the unfermented CRPP were 10.77 ± 0.27 mg GAE/g, 4.78 ± 0.07 mg QE/g, 22.19 ± 0.97, and 13.35 ± 0.71 µmol TE/g, respectively. As pH increased from 4.0 to 6.5, the TPC, TFC, and ABTS scavenging capacity showed a decreasing trend, however, DPPH scavenging capacity showed no significant change. At pH 4.0, the TPC and ABTS scavenging capacity of the fermented CRPP significantly (*p* < 0.05) increased to 11.97 ± 0.26 mg GAE/g and 25.43 ± 1.89 µmol TE/g, respectively, while TFC and DPPH scavenging capacity slightly (*p* > 0.05) increased to 4.83 ± 0.11 mg QE/g and 14.05 ± 1.12 µmol TE/g, respectively. 

We further investigated the changes in the phytochemicals in CRPP fermented with different pH. Figure 1 shows the HPLC chromatographs of the standard and CRPP samples and Table 2 presents the quantified contents in different samples. The results show that chlorogenic acid, caffeic acid, p-coumaric acid, and naringenin (compounds **1**–**3**, **7**) are not able to be quantified in CRPP, while the other six compounds including ferulic acid, narirutin, hesperidin, hesperetin, nobiletin, and tangeretin (compounds **4**–**6**, **8**–**10**) were identified and quantified. Hesperidin is the most abundant flavonoid in CRPP with a concentration of 19.36 ± 0.47 mg/g, followed by nobiletin (6.31 ± 0.11 mg/g), narirutin (4.97 ± 0.07 mg/g), and tangeretin (2.91 ± 0.04 mg/g). The contents of ferulic acid and hesperetin are relatively low. After fermentation at pH 4.0, narirutin content significantly (*p* < 0.05) increased to 5.53 ± 0.13 mg/g, while the other five compounds remained unchanged. Further increasing the pH value to 6, the contents of all six compounds either decreased or remained consistent. According to the results of TPC, TFC, antioxidant activity, and phytochemical composition, pH 4 was selected for the following fermentation experiments. 

### 2.2. Effect of Fermentation Temperature on TPC, TFC, Antioxidant Activity, and Phytochemical Compositions in CRPP 

Under pH 4.0, the CRPP was fermented at temperatures of 25, 30, and 35 °C, respectively. The results show that ABTS and DPPH scavenging capacity did not change significantly when the incubation temperature increased from 25 to 35 °C (Table 1). Increasing the temperature from 25 to 35 °C caused a significant decrease in TFC from 4.71 ± 0.18 µmol TE/g to 4.01 ± 0.14 µmol TE/g, while TPC significantly increased from 11.41 ± 0.35 µmol TE/g to 12.05 ± 0.44 µmol TE/g. When incubated at 25 and 35 °C, CRPP also had lower hesperidin, nobiletin, and tangeretin contents than the unfermented CRPP (Table 2). Therefore, in the further experiments, the fermentation temperature was set at 30 °C. 

### 2.3. Effect of Moisture Content on TPC, TFC, Antioxidant Activity, and Phytochemical Compositions in CRPP 

Table 1 shows that increasing the moisture content (MC) from 70% to 80% resulted in a significant (*p* < 0.05) increase in TPC from 11.97 ± 0.26 to 12.95 ± 0.59 mg GAE/g, while TFC, and ABTS and DPPH scavenging capacities did not increase significantly (*p* > 0.05). However, TFC, and ABTS and DPPH scavenging capacities decreased as the MC further increased to 90%. Similarly, ferulic acid and hesperidin also significantly increased when the MC increased from 70 to 80% and then decreased as the MC reached 90%. At the MC of 90%, nobiletin and tangeretin also showed a decreasing trend. Therefore, the MC of 80% was selected for the following experiments. 

### 2.4. Effect of Spore Concentration on TPC, TFC, Antioxidant Activity, and Phytochemical Compositions in CRPP 

Spore concentrations ranging from 4 × 10^6^ to 4 × 10^7^ spores/g CRPP were used to inoculate CRPP. Compared to the control, TPC, TFC, and ABTS and DPPH scavenging capacities increased significantly (*p* < 0.05) in the CRPP after fermentation with different spore concentrations of *A. niger* CGMCC 3.6189 and showed an increasing trend when the inoculum concentration increased from 4 × 10^6^ to 4 × 10^7^ spores/g (Table 1). At the spore concentration of 4 × 10^7^ spores/g, TPC, TFC, and ABTS and DPPH scavenging capacities increased to 13.47 ± 0.36 mg GAE/g, 5.15 ± 0.23 mg QE/g, 27.53 ± 0.24 µmol TE/g, and 17.33 ± 1.40 µmol TE/g, respectively. The contents of hesperidin, nobiletin, and tangeretin significantly increased with the increased spore concentrations, however, narirutin showed a significant decrease when the spore concentration increased from 4 × 10^6^ to 4 × 10^7^. The spore concentration of 4 × 10^7^ spores/g was selected for the further fermentation.

### 2.5. Effect of Fermentation Times on TPC, TFC, Antioxidant Activity, and Phytochemical Compositions in CRPP 

Increasing the fermentation time resulted in significant increases (*p* < 0.05) in TPC, TFC, and ABTS and DPPH scavenging capacities, which reached 18.31 ± 0.35 mg GAE/mg, 6.06 ± 0.11 mg QE/g, 36.60 ± 1.82 µmol TE/g, and 22.91 ± 3.45 µmol TE/g, respectively, when CRPP was fermented for 192 h. Hesperidin also reached the maximum of 28.23 ± 0.76 mg/g after 192 h fermentation, however, nobiletin and tangeretin contents were highest with 96 h fermentation. According to the experimental results, the optimal SSF conditions for flavonoid accumulation and antioxidant activity improvement in CRPP were pH 4.0, moisture content 80%, temperature 30 °C, inoculum concentration 4 × 10^7^ spores/g, and fermentation time 192 h. 

### 2.6. Effect of Fermentation Conditions on the Color of CRPP

Alongside the chemical and bioactivity changes in CRPP after fermentation, we also assessed the impact of the fermentation conditions on the physical change of CRPP. Figure 2a shows the changes in the color indexes of CRPP after fermentation under different conditions. The results show that L*, a*, and b* values of CRPP significantly decreased with different fermentation conditions compared with the unfermented CRPP, while the CCI values significantly increased. Among all these five factors, fermentation time was the most important factor that affected the changes in the color parameters, while the pH of the inoculum had the least effect. Increasing fermentation time significantly decreased L*, a*, and b* values while increasing CCI and ∆E* values. Within the first 96 h fermentation, the changes in the color indexes were mild, however, these color indexes changed dramatically with the extended fermentation times, thereby producing much darker CRPP samples. It is noted that color is also an important attribute that influences consumers’ choices. Although increasing the fermentation time to 192 h significantly enhanced the accumulation of flavonoids and the antioxidant activity of CRPP, the extended fermentation time tends to produce CRPP with an unpleasantly dark color (Figure 2b). Therefore, if considering both the bioactivity and the organoleptic quality of CRPPs, a fermentation time of 96 h should be chosen.

### 2.7. Pearson Correlation between SSF Conditions and Quality Attributes of CRPP 

The Pearson correlation analysis was used to describe the relationship between the TPC, TFC, antioxidant activities (ABTS and DPPH), phytochemical compositions, color parameters, and fermentation conditions including pH, spore concentration, moisture content, incubation temperature, and the fermentation time. Figure 3 shows that the contents of nobiletin, tangeretin, and hesperidin, the TPC, TFC, ABTS and DPPH scavenging capacities, ∆E*, and CCI were positively correlated (*p* < 0.05) with the fermentation time and spore concentration, but were not correlated to the fermentation temperature. The contents of nobiletin and hesperidin and the TPC, TFC, and ABTS and DPPH scavenging capacities were also positively (*p* < 0.05) correlated with the initial moisture content. Moreover, the contents of hesperidin, nobiletin, and tangeretin, the TPC, and TFC were positively (*p* < 0.05) correlated with ABTS and DPPH scavenging capacities of CRPP, suggesting that these flavonoids and phenolic compounds are closely associated with the antioxidant activity of citrus peel. The ∆E* and CCI showed a positive (*p* < 0.05) correlation with the contents of nobiletin and tangeretin, TPC, TFC, and ABTS and DPPH scavenging capacities, while the color indexes L*, a*, and b* showed negative correlation to these six attributes, indicating that color parameters can be good indicators for assessing the antioxidant activity of citrus peel that has undergone fermentation. The pH was also negatively correlated with the contents of ferulic acid and hesperidin. Our results suggested that fermentation temperature and spore concentration are two of the most important factors that affect the phenolic and flavonoid contents, the antioxidant activity, as well as the color of CRPP during SSF, while moisture content has a significant impact on the bioactive components and bioactivity, but not the color of CRPP. Fermentation time and initial pH have much less influence on the quality attributes and the organoleptic characteristics of CRPP. We also found that the flavonoids such as hesperidin, nobiletin, and tangeretin as well as the color indexes (L*, a*, b*, ∆E*, and CCI) are good chemical and physical indicators for the fermented CRPP with high antioxidant activity.

### 2.8. Principal Component Analysis of the Quality Attributes of CRPP 

We performed further principal component analysis (PCA) to visualize the relationship between the observations and variables. A total of twenty samples including nineteen fermented and one unfermented sample were analyzed, and the results are shown in Figure 4. The first and second PC contribute to 67.44% and 15.64% of the total variance, respectively. The result shows that FC1a, Fc2a, FC2b, FC2c, FC3a, FC3b, FC4a, FC4b, FC4c, and FC5a were more similar to the control and were characterized by similar L*, a*, and b* values. The samples including FC1b, FC1c, FC1d, FC1e, FC1f, FC2a, and FC3c were identified by low ferulic acid, narirutin, and hesperidin contents, while FC5b, FC5c, and FC5d were distinguished by high contents of TPC, TFC, nobiletin, and tangeretin, ABTS and DPPH scavenging capacities, ∆E*, and CCI, especially for the FC5d CRPP. 

## 3. Discussion

SSF is a low-moisture fermentation technique that has been used feasibly and economically for large-scale bioconversion and biodegradation of agri-food waste or by-products [15]. Fermentation conditions such as pH, temperature, moisture content, microbial concentration, and fermentation time are critical factors that affect microbial growth during SSF, thus influencing the chemical, biological, and organoleptic qualities of the products. pH is a determining factor for the growth of microorganisms due to its influence on enzyme activity, cellular processes, and complex physiological phenomena such as membrane permeability and morphology [16]. Previous research showed that *A. niger* grown at a pH ranging from 5 to 6 accelerated the accumulation of flavonoids [17,18]. Initial media pH values between 6.5 and 7.5 were optimal for flavonoid accumulation by *A. niger* in *Isatis tinctoria* L. hairy root [19]. Using *A. niger* B1b, Ahmed et al. [20] found the optimalpH of 8.5 for phenolic accumulation. Since it is hard to measure the pH of the solid substrate in the SSF, the pH of the initial inoculum was used to assess the effect of pH on the accumulation of flavonoids. In the present study, we found that the inoculum with a pH of 4.0 was the best for flavonoid accumulation, which may be related to the effect of pH on enzyme activity. pH has been shown to have a direct effect on the activity of enzymes. pH influences the ionization of the components in the growth media and the synthesis of enzymes [21]. Rutinosidase, naringinase, hesperidinase, a-L-rhamnosidase, pectinase, cellulase, tannase, phytase, β-glucosidase, and lipase have been used in citrus products flavonoid biotransformation under fermentation conditions of pH from 3.5 to 7.5, temperature from 30 to 70 °C, and incubation time from 2 to 120 h [6]. A study of the effects of pH on the phenolics, flavonoids, and antifungal activity in the liquid culture medium fermented with *A. tamarii* revealed that pH 5 was ideal [16]. They also showed that strong acidic (pH = 3), neutral, and basic (pH = 9 and 11) conditions significantly decreased the TPC and TFC. However, the starting pH of 7 was the best to accumulate flavonoids in the root of *Isatis tinctoria* L. fermented by immobilized *A. niger* 3.3883 [19]. In contrast to our study, the root of *Isatis tinctoria* L. was submerged in the liquid culture media and *A. niger* was immobilized in Ca-alginate gel beads. Therefore, it is important to screen the optimal pH for the accumulation of flavonoids in the fermentation systems with different conditions, substrates, and microorganisms.

Fermentation temperature affects the heat and mass transfer as well as microbial growth and activity. Low temperature limits microbial growth and production of bioactive compounds, while high temperature also disturbs the growth of or even kills the microbes, and, thus, inhibits the formation of products. The temperature tolerance of *A. niger* isolated from the Himalayan soil was in a range of 9–42 °C, with an optimal growing temperature at 28 °C [22]. We also found that the optimal temperature for *A. niger* to accumulate the phenolics and flavonoids in citrus peel was 30 °C. A similar result was found by Jiao et al. [19], who observed the highest flavonoid production in the roots of *Isatis tinctoria* L. fermented by immobilized *A. niger* at 30 °C. Another optimal temperature for flavonoid accumulation was also reported. For example, Bose et al. [16] found that *A. tamarii* grown at 35 °C produced the highest TPC and TFC, while phenolics and flavonoids were produced at a much lower level when grown at 15 and 45 °C. Liu et al. [23] also reported that the optimal temperature to accumulate flavonoids in dandelion during SSF was 35 °C. The optimal temperature for flavonoid accumulation may be due to the different substrates and microorganisms used. 

During fermentation, the substrates must contain enough moisture to enable microbial development [24]. In SSF, the moisture content of the substrate usually ranges from 30 to 85% [25]. The heat applied and produced in SSF causes the low-moisture sample to dry out, resulting in the poor growth of microorganisms [15]. Low moisture content also reduces the solubility of nutrients in the substrate, causing reduced availability of nutrients for microbial growth [24]. However, high moisture reduces the porosity of the solid matrix and leads to the aggregation of substrate particles, thereby limiting the oxygen transfer [24], and thus inhibiting microbial growth. Therefore, to maximize the growth of microbes and the accumulation of flavonoids, appropriate moisture content needs to be selected. The initial moisture contents of 60–90% were used to ferment citrus by-products to produce multi-enzymes using different fungi and the results showed that the optimal moisture content for *A. niger* BTL was 90% [24]. In our study, we found that phenolics and flavonoid accumulation followed a trend of first increasing and then decreasing as the moisture content increased from 70 to 90%, with a maximum accumulation at the initial moisture content of 80%. A similar trend was also observed in citric acid production in citrus peel using *A. niger* CECT-2090 [26]. For the fermentation of dandelion by the mixture of *L. plantarum* and *S. cerevisiae* in solid state, a moisture content of about 53% was best to accumulate the flavonoids [23]. Inoculum concentration is an important factor that promotes microbial growth and metabolite production in SSF. Flavonoid content increased when the spore concentration of immobilized *A. niger* was increased from 10 to 10^4^ spore/mL [19]. Liu et al. [23] reported that as the inoculum concentration increased, the flavonoid content first increased and then decreased, reaching a maximum at the inoculum concentration of 1.2 × 10^7^ spores/g. A concentration of 2.5 × 10^5^ spores/g of *A. niger* 3.13901 improved the flavonoid accumulation in *Citrus reticulata* peel [27]. Cai et al. [28] used 10^6^ spores/g *A. oryzae and A. niger* to increase the TPC and TFC in fermented oats. We found that the highest accumulation of flavonoids and phenolics was in the CRPPs inoculated with 4 × 10^7^ spores/g. It is known that the increase in the inoculum concentration can shorten the fermentation time and limit the growth of other microorganisms [29]. However, high inoculum level increased the crowdedness of the microorganisms, leading to the enhanced consumption of sugar, and thus resulting in the reduction in bioactive productivity [30].

Fermentation time is also a critical factor that affects the TPC, TFC, antioxidant activity, and the biotransformation of flavonoids in citrus peels. The fermentation time is determined by the nature of the medium, the fermenting organisms, the concentration of nutrients, and the physiological parameters of the process [18]. Ahmed et al. found that the phenolic compounds were the highest with fermentation of *A. niger* B1b for 9 days [20]. Pérez-Nájera et al. [31] showed that TPC, TFC, and antioxidant activity of lime peels fermented by *A. saitoi* remained unchanged within the first 5 days of fermentation, while significantly increased to 8.66, 5.14, and 5.8 times, respectively, after 6 days fermentation, followed by a dramatic decrease when fermentation time extended to 7 days. The maximum hesperidin content was observed in the lime peels fermented for 2 days. The much higher increases in the TPC, TFC, and antioxidant activity of lime peels compared to our results may be due to the different fermentation conditions, substrates, and microorganisms used. We found that fermentation time was the most important factor that affects the accumulation of phenolics and flavonoid compositions in citrus peels. TPC, TFC, and antioxidant activity significantly increased as the fermentation time extended. A similar result was reported by Liu et al. [23], who found that fermentation time was the only factor that significantly affect the flavonoid content in dandelion fermented by a mixture of *L. plantarum* and *S. cerevisiae* using a four-factor response surface methodology design. They found the maximum flavonoid content was obtained when dandelion was fermented for 52 h. Metabolomics analysis further showed that in the fermented dandelion, 27 flavonoids were upregulated and 30 flavonoids were downregulated. Santos et al. [32] showed that TPC peaked at 48 h of fermentation; however, the TFC did not reach this peak even at 168 h of fermentation in *Passiflora ligularis* seed. We also found that the times for obtaining the maximum phenolic and flavonoid contents were different. For example, the ferulic acid and narirutin reached their maximum at a fermentation time of 144 h, hesperidin peaked at 192 h, while nobiletin and tangeretin were the highest after 96 h fermentation. This result may be related to the different enzyme activities that are needed for the biosynthesis of flavonoids. 

The increases in TPC, TFC, and antioxidant activity in the fermented CRPP can be attributed to the enzymes involved in the biosynthesis of flavonoids, as well as the hydrolases. In plants, polyphenolic compounds exist in bounded and free forms. Cellulases, xylanases, and ligninases can release bounded polyphenolic compounds from the cell wall through disruption of the hemicellulose, cellulose, and lignin, thus increasing the free phenolic compounds. β-glucosidase hydrolyzes phenolic glycosides to release free phenolics and tannases catalyze the breaking of ester bonds and depside linkage of the polyphenol complexes to produce smaller phenolic compounds with higher antioxidant activity [32]. Moreover, other enzymatic reactions such as hydroxylation, dihydroxylation, dehydrogenation, methylation, oxidation, and reduction reactions occurring during microbial fermentation may also contribute to the increased antioxidant activity of citrus peels after fermentation due to the production of compounds with higher antioxidant activity [33,34]. Cyclization of chalcones or transformation of other compounds can also increase flavonoid accumulation and antioxidant activity [35]. It is well known that antioxidant activity is closely associated with the phenolics and flavonoids in the plant extract. We found that ABTS and DPPH scavenging capacities of fermented CRPP were positively correlated with the TPC, TFC, and contents of hesperidin, hesperetin, and nobiletin. Similar results were found by Long et al. [36] and Guo et al. [37], who also reported a positive correlation between ABTS and DPPH scavenging ability of *Citrus sinensis* extract or citrus peel extract and TPC, TFC, and nobiletin content.

Aside from the nutritional value of food products, color is another important quality attribute that influences the acceptability of foods. The color of food products is closely associated with the physical, chemical, biochemical, and microbial reactions during the postharvest storage or processing of food products [38]. Therefore, color changes can be used to predict the changes in other quality attributes of food products. The color indexes a^*^ (redness (+) or greenness (–), b^*^ (yellowness (+) or blueness (–), L^*^ (brightness (100) or darkness (0), and ΔE (the total color difference) are generally used to assess products’ changes of color quantitatively [39]. The citrus color index (CCI) is specifically used to measure the variable of color parameters of citrus products and by-products. CCI ≤ −5 indicates dark green color, −5 < CCI ≤ 0 indicates green color, 0 < CCI ≤ 3 corresponds to yellowish green color, 3 < CCI ≤ 6 indicates greenish yellow color, 6 < CCI ≤ 8 represents yellowish orange color, 8 < CCI ≤ 10 indicates orange color, and CCI > 10 corresponds to dark orange color [40]. Similar to the changes in the TPC, TFC, antioxidant activity, and flavonoid compositions in CRPP during SSF, the color of CRPP was also significantly affected by the fermentation conditions, among which fermentation time was also the most significant influencing factor. Compared with the unfermented CRPP, the fermented CRPP has lower a^*^, b^*^, and L^*^ values, indicating that the redness, yellowness, and brightness of CRPP decreased after fermentation. Similar results were also observed in the tempe, a nutritious food prepared from the fermentation of soybeans by *Rhizopus* spp. [38]. One possible explanation for this is that high fermentation temperatures promote fungal growth, resulting in an early formation of spores and affecting the color of the fermented product. For example, black spores of *A. niger* were only observed after 72 h at 34 °C and after 96 h at 31 °C, but not after 120 h for 25 and 28 °C [41]. Moreover, high temperatures can accelerate the degradation of chlorophyll, causing caramelization and the Maillard reaction, which produce browning [14]. 

## 4. Materials and Methods

### 4.1. Materials and Chemical

The fruits of *Citrus Reticulate* Blanco ‘Chachiennsis’ (Chachi) were harvested from an orchard in Xinhui, Guangdong Province, China, on 8 November 2020. The peels were collected and sun-dried for 5 days followed by vacuum sealing in plastic bags. The samples were stored in a desiccator at room temperature. *Aspergillus niger* CGMCC 3.6189 was purchased from China General Microbiological Culture Collection Center (Beijing, China). The Folin–Ciocalteu, potato dextrose agar (PDA), and yeast powder were provided by Solarbio^®^ Science and Technology Co., Ltd. (Beijing, China). The ABTS and 1, 1-DPPH were bought from Shanghai Macklin Biochemical Co., Ltd. (Shanghai, China). The HPLC standards (chlorogenic acid, caffeic acid, *p*-coumaric acid, ferulic acid, narirutin, hesperidin, naringenin, hesperetin, nobiletin, and tangeretin) were purchased from Chengdu-Must Technology Co., Ltd. (Chengdu, China). The other chemicals were all of analytical grades and were acquired from Sinopharm Chemical Reagent Co., Ltd. (Zhenjiang, China).

### 4.2. Samples Preparation

The peels of the “Chachi” fruits were sun-dried until the moisture content reached 11% (w.b., wet basis). The peels were ground into fine powders and sieved through a 50-mesh stainless steel sieve. The *Citrus Reticulate* peel powders (CRPP) were stored at 4 °C for the following experiments.

### 4.3. Preparation of the Growth Curve and the Inoculum of Aspergillus niger CGMCC 3.6189

*Aspergillus niger* CGMCC 3.6189 grown on the PDA (pH 5.6 ± 0.2) was inoculated in a culture medium containing 10 g/L glucose and 20 g/L yeast extract, and the initial concentration of the inoculum was adjusted to OD 600 of 0.1. The inoculum was grown in an incubator (LHS-100CL, Shanghai Yiheng Technology Co. Ltd., Shanghai, China) at 30 °C and 100% relative humidity (RH). The spore suspension was taken every hour until 32 h. The absorbance at 600 nm was recorded. The concentration of the spore solution was determined using the method described by Bastidas [42]. Briefly, 0.5 mL of 10^6^ diluted spore solutions was mixed with 0.5 mL of methylene blue (1%) and then 10 µL of the mixture was read under a microscope using a hemocytometer. The concentration of the spores was calculated using the following equation: (1)$$Spore\;concentration=\frac{Number\;of\;cells\times 10,000}{Number\;of\;square\times times\;of\;dilution}$$

The growth curve was plotted using the culture time and the logarithm of the spore number. The spores grown at mid-log phase (after 15 h growing in the culture medium) were used to inoculate CRPP. 

### 4.4. Solid-State Fermentation (SSF) 

SSF was conducted using the method described by Wang et al. [7] with minor modifications. Briefly, 1.5 g of CRPP was placed in a Petri dish and sterilized for 30 min on each side before the addition of the mid-log phase spores’ suspension, followed by incubation in an incubator (LHS-100CL, Shanghai Yiheng Technology Co. Ltd., Shanghai, China) under 100% RH for different times. The experimental factors included pH (4.0, 4.5, 5.0, 5.5, 6.0, and 6.5), temperature (25, 30, and 35 °C), moisture content (70, 80, and 90% w.b.), inoculum concentration (4 × 10^6^, 2 × 10^7^, and 4 × 10^7^ spores of *A. niger*/g of CRPP), and fermentation time (60, 96, 144, and 192 h).

### 4.5. Extraction of CRPP

The CRPP was extracted using a modified ultrasound-assisted method described by Luo et al. [43]. Briefly, the unfermented (control) and fermented samples were freeze-dried and extracted with 80% methanol at a solid-to-solvent ratio of 1: 30 (*w*/*v*) for 20 min under ultrasonication. The extract was centrifuged at 5000 rpm, 4 °C for 20 min, filtered through a 0.22 μm filter membrane, and the supernatant was stored at 4 °C for further analyses.

### 4.6. Analysis of Total Phenolic Content (TPC)

The TPC of CRPP extracts was determined using the Folin–Ciocalteu method described by Chen et al. [44]. Briefly, 20 µL of diluted CRPP extracts or the standard solution (0.1 mg/mL gallic acid) were mixed with 100 µL of Folin–Ciocalteu solution (10 times diluted) and incubated in darkness for 1 min, followed by the addition of 80 µL of Na_2_CO_3_ (75 mg/mL) and further incubation for 30 min. Absorbance was measured at 765 nm using a Spark^®^ 10M multimode microplate reader (Tecan, MA, USA). The results were expressed as mg Gallic acid equivalents (GAE)/g of CRPP (d.w).

### 4.7. Analysis of Total Flavonoid Content (TFC)

The TFC of CRPP extracts was determined using a spectrophotometric method according to Shraim et al. [45] with some modifications. Briefly, in a 15 mL glass test tube, 1000 µL appropriated diluted CRPP extracts or quercetin standard solution (0.2 mg/mL) was mixed with 60 µL NaNO_2_ (5%). The mixture was allowed to stay in the dark for 5 min. Thereafter, 60 µL AlCl_3_ (10%) was added, followed by the addition of 400 µL NaOH (1.0 mol/L). After 6 min incubation in darkness, all the solutions were vortexed and the absorbance was recorded at 510 nm against methanol 80% as blank using a 96-well microplate reader (Tecan, MA, USA). The results were expressed as mg quercetin equivalents (QE)/g of CRPP (d.w.).

### 4.8. Analysis of ABTS Radical Scavenging Capacity

The ABTS scavenging capacity was evaluated according to Chen et al. [46], with slight modifications. Briefly, 20 µL of CRPP extract or Trolox standard (0.25 mmol/L) was reacted with 180 µL of ABTS working solution. After 10 min incubation in darkness, the absorbance intensity was measured at 734 nm. The results were expressed in µmol Trolox equivalents (TE)/g CRPP (d.w.).

### 4.9. Analysis of DPPH Radical Scavenging Capacity

The DPPH scavenging capacity was assessed according to the method described by Chen et al. [46], with minor modifications. An aliquot of 180 µL of methanol-diluted CRPP extract or Trolox standard (0.2 mmol/L) was reacted with 20 µL of DPPH reagent (0.394 g/L). After 10 min incubation in darkness, the absorbance was read at 519 nm. The DPPH radical scavenging capacity was expressed as µmol Trolox equivalents (TE)/g of CRPP (d.w.).

### 4.10. Analysis of Phytochemicals Using HPLC 

The phytochemicals in CRPP extracts were determined using HPLC according to the method of Gao et al. [47]. An LC-20AD HPLC instrument (Shimadzu L.C., Kyoto, Japan) equipped with a diode array detector, and a Phenomenex Kinetex C18 (100 × 4.8 mm, 5 μm) column (Phenomenex, Torrance, CA, USA) were used and the temperature of the column was set at 25 °C. The CRPP extract was eluted with 0.1% TFA (solvent A) and acetonitrile (solvent B) at a flow rate of 1.0 mL/min. The elution gradient includes: 0–5 min, 15–20% B; 5 −10 min, 20% B; 10 −16 min, 20–25% B; 16−17 min, 25–26% B; 17–25 min, 26–27% B; 25–28 min, 27–30% B; 28–33 min, 30–40% B; 33–40 min, 40–65% B; 40–45 min, 65–15% B; and 45–50min, 15% B. Chlorogenic acid, caffeic acid, *p*-coumaric acid, ferulic acid, nobiletin, and tangeretin were detected at 330 nm. Narirutin, hesperidin, hesperetin, and naringenin were detected at 280 nm. The phytochemical content was expressed as mg/g of CRPP (d.w.).

### 4.11. Determination of Color Parameters

The color parameters (L*, a*, and b*) of the CRPP were evaluated using a colorimeter (WS-2300, iWAVE Co. Ltd., Zibo, China). The citrus color index (CCI) value and the total color difference (ΔE*) were determined according to Arzam et al. [41] and Sun et al. [40] using the following equations: (2)$$\mathrm{CCI}=\frac{100\times a*}{L*\times b*}$$
(3)$${\Delta E\;}^{*}=\sqrt{{\left(a^{*}-a_{0}^{*}\right)}^{2}+{\left(b^{*}-b_{0}^{*}\right)}^{2}+{\left(L^{*}-L_{0}^{*}\right)}^{2}}$$
where L*, a*, and b* are the color index of the fermented CRPP and L_0_*, a_0_*, and b_0_* are the color index of the unfermented CRPP (control).

### 4.12. Statistical Analysis

The data are expressed as mean ± SD. One-way ANOVA with Tukey’s test was used to evaluate the significant differences between CRPP samples using MINITAB 18 (Minitab Ltd., State College, PA, USA). A *p* < 0.05 represents a significant difference. Pearson correlation analysis and principal component analysis (PCA) were performed using Origin 9.9 software (OriginLab Co., Northampton, MA, USA).

## 5. Conclusions

In the present study, we found that *A. niger* CGMCC 3.6189 can increase the phenolic and flavonoid contents and the antioxidant activity of citrus peel in SSF when the fermentation conditions are appropriately controlled. Hesperidin, nobiletin, narirutin, and tangeretin are four of the major flavonoids in CRPPs, all of which were significantly increased after SSF by *A. niger*. The maximum flavonoid accumulation conditions were pH 4.0, temperature 30 °C, moisture content 80%, and spore concentration 4 × 10^7^ spores/g d.w. for 192 h. Among these five factors, fermentation time, spore concentration, and moisture content are the three most important factors that affect flavonoid accumulation, antioxidant activity, and the color of CRPP, while the fermentation temperature has the least impact. Although long-time fermentation significantly increased the flavonoid contents and antioxidant activity, it also caused the production of CRPP with a much darker color. Therefore, in consideration of both bioactive components and the organoleptic characteristics of CRPP, a fermentation time of 96 h is a better choice. Thus, we recommended fermentation by *A. niger* CGMCC 3.6189 as an alternative method for bioactive compound accumulation in *Citrus reticulata* peel. Nevertheless, future investigations on energy source effects are needed to reduce the fermentation time that affected the color of the peel powder and to help in the optimization of the flavonoid accumulation.

## Figures and Tables

**Figure 1 molecules-27-08949-f001:**
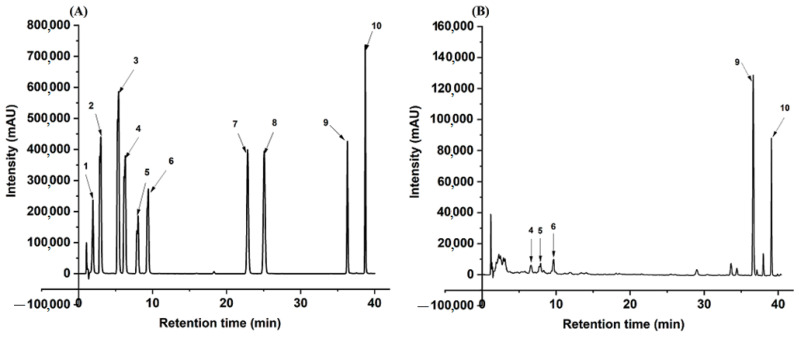
HPLC profile of *Citrus reticulata* peels powder extract at 330 nm. Note: HPLC chromatographs of mixed standard (**A**) and the extract of *Citrus reticulata* peel fermented for eight days (**B**). The numbers of compounds from 1 to 10 correspond to the tested phenolic compounds: **^1^** chlorogenic acid; **^2^** caffeic acid, **^3^** P-coumaric acid, **^4^** ferulic acid, **^5^** narirutin, **^6^** hesperidin, **^7^** naringenin, **^8^** hesperetin, **^9^** nobiletin, **^10^** tangeretin.

**Figure 2 molecules-27-08949-f002:**
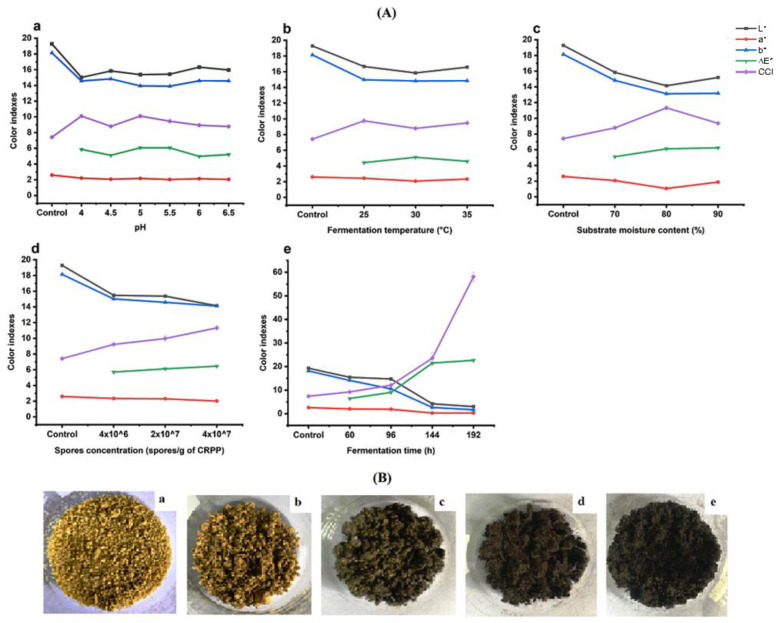
(**A**) The changes in the color parameters (L*, a*, b*, ∆E*, and CCI) of Citrus reticulata peels under different fermentation conditions: a = pH, b = fermentation temperature, c = moisture content, d = spore concentration, e = fermentation time. (**B**) The images of CRPP: a = unfermented CRPP, b = 60 h fermentation, c = 96 h fermentation, d = 144 h fermentation, e = 192 h fermentation.

**Figure 3 molecules-27-08949-f003:**
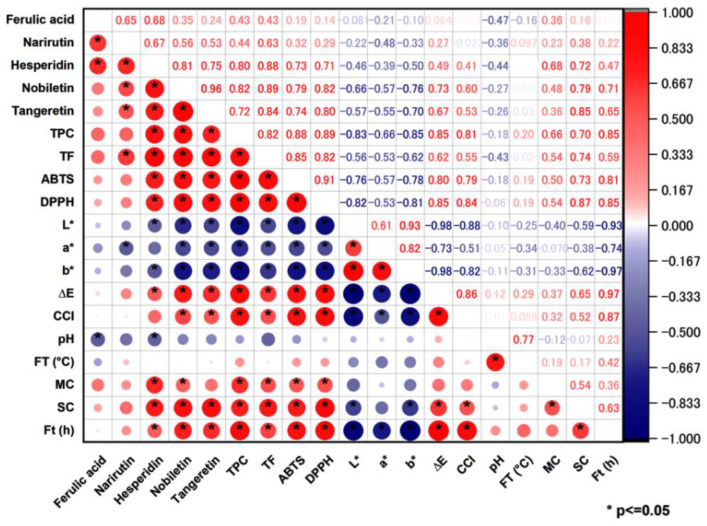
Pearson correlation between the fermentation conditions and the quality attributes of Citrus reticulata peel under solid-state fermentation. **Note**: CCI = citrus color index, FT = fermentation temperature, MC = moisture content, SC = spore concentration, Ft = fermentation times. * Asterisk denotes significant difference at *p* < 0.05.

**Figure 4 molecules-27-08949-f004:**
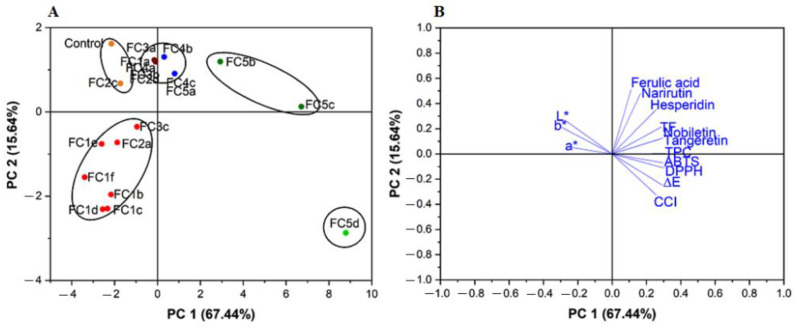
Principal component analysis of Citrus reticulata peels powder under Aspergillus niger GCMCC 3.6189 solid-state fermentation: (**A**) Score plots (**B**) loading plot. Table 1 and Table 2 contain a detailed description of the sample names.

**Table 1 molecules-27-08949-t001:** Effects of solid-state fermentation conditions on the TPC, TFC, and ABTS and DPPH scavenging capacity of *Citrus reticulata* peel.

Sample Name	Variables	TPC (mg GAE/g)	TFC (mg QE/g)	ABTS (µmol TE/g)	DPPH (µmol TE/g)
pH ^1^					
Control	-	10.77 ± 0.27 ^b^	4.78 ± 0.07 ^a^	22.19 ± 0.97 ^abc^	13.35 ± 0.71 ^a^
FC1a	4.0	11.97 ± 0.26 ^a^	4.83 ± 0.11 ^a^	25.43 ± 1.89 ^a^	14.05 ± 1.12 ^a^
FC1b	4.5	10.57 ± 0.18 ^b^	4.28 ± 0.15 ^b^	25.07 ± 1.09 ^ab^	14.65 ± 0.49 ^a^
FC1c	5.0	10.40 ± 0.08 ^b^	4.06 ± 0.17 ^b^	24.97 ± 1.99 ^ab^	14.61 ± 0.82 ^a^
FC1d	5.5	10.48 ± 0.09 ^b^	4.07 ± 0.09 ^b^	23.63 ± 0.20 ^ab^	13.56 ± 0.46 ^a^
FC1e	6.0	10.41 ± 0.30 ^b^	3.67 ± 0.10 ^c^	21.46 ± 1.91 ^bc^	13.65 ± 0.90 ^a^
FC1f	6.5	10.44 ± 0.36 ^b^	3.51 ± 0.06 ^c^	18.72 ± 0.17 ^c^	13.61 ± 0.92 ^a^
Incubation temperature (IT, °C) ^2^					
Control	-	10.77 ± 0.27 ^b^	4.78 ± 0.07 ^a^	22.19 ± 0.97 ^a^	13.35 ± 0.71 ^a^
FC2a	25	11.41 ± 0.35 ^ab^	4.71 ± 0.18 ^a^	24.56 ± 1.69 ^a^	13.44 ± 0.69 ^a^
FC2b	30	11.97 ± 0.26 ^a^	4.83 ± 0.11 ^a^	25.43 ± 1.89 ^a^	14.05 ± 1.12 ^a^
FC2c	35	12.05 ± 0.44 ^a^	4.01 ± 0.14 ^b^	22.49 ± 1.20 ^a^	13.33 ± 1.44 ^a^
Moisture content (MC,%, w.b.) ^3^					
Control	-	10.77 ± 0.27 ^c^	4.78 ± 0.07 ^a^	22.19 ± 0.97 ^b^	13.35 ± 0.71 ^a^
FC3a	70	11.97 ± 0.26 ^b^	4.83 ± 0.11 ^a^	25.43 ± 1.89 ^a^	14.05 ± 1.12 ^a^
FC3b	80	12.95 ± 0.59 ^a^	4.90 ± 0.06 ^a^	25.63 ± 0.47 ^a^	15.00 ± 1.41 ^a^
FC3c	90	13.43 ± 0.20 ^a^	4.29 ± 0.10 ^b^	24.73 ± 0.45 ^ab^	14.08 ± 0.44 ^a^
Spore concentration (SC, spores/g) ^4^					
Control	-	10.77 ± 0.27 ^b^	4.78 ± 0.07 ^b^	22.19 ± 0.97 ^c^	13.35 ± 0.71 ^b^
FC4a	4 × 10^6^	12.95 ± 0.59 ^a^	4.90 ± 0.06 ^ab^	25.63 ± 0.47 ^b^	15.00 ± 1.41 ^ab^
FC4b	2 × 10^7^	13.17 ± 0.40 ^a^	5.04 ± 0.04 ^ab^	26.00 ± 0.21 ^b^	16.29 ± 1.83 ^ab^
FC4c	4 × 10^7^	13.47 ± 0.36 ^b^	5.15 ± 0.23 ^a^	27.53 ± 0.24 ^a^	17.33 ± 1.40 ^a^
Fermentation time (FT, h) ^5^					
Control	-	10.77 ± 0.27 ^d^	4.78 ± 0.07 ^d^	22.19 ± 0.97 ^c^	13.35 ± 0.71 ^c^
FC5a	60	13.47 ± 0.36 ^c^	5.15 ± 0.23 ^c^	27.53 ± 0.24 ^b^	17.33 ± 1.40 ^bc^
FC5b	96	13.77 ± 0.21 ^c^	5.35 ± 0.20 ^bc^	28.13 ± 1.36 ^b^	17.36 ± 1.09 ^bc^
FC5c	144	15.69 ± 0.33 ^b^	5.69 ± 0.21 ^ab^	29.37 ± 0.52 ^b^	18.99 ± 1.55 ^ab^
FC5d	192	18.31 ± 0.35 ^a^	6.06 ± 0.11 ^a^	36.60 ± 1.82 ^a^	22.91 ± 3.45 ^a^

Note: Means that do not share a letter are significantly different; data are expressed as mean ± SD (*n* = 3) on a dry weight basis. ^1^ The SSF conditions are as follows: incubation temperature = 30 °C, moisture content = 40%, spore concentration = 4 × 10^6^ spores/g, fermentation time = 60 h, pH = 4.0, 4.5, 5.0, 5.5, and 6.0, respectively. ^2^ The SSF conditions are as follows: moisture content = 40%, spore concentration = 4 × 10^6^ spores/g, fermentation time = 60 h, pH = 4.0, incubation temperature = 25, 30, and 35 °C, respectively. ^3^ The SSF conditions are as follows: spore concentration = 4 × 10^6^ spores/g, fermentation time = 60 h, pH = 4.0, incubation temperature = 30 °C, moisture content = 70, 80, and 90%, respectively. ^4^ The SSF conditions are as follows: moisture content = 80%, fermentation time = 60 h, pH = 4.0, incubation temperature = 30 °C, spore concentration = 4 × 10^6^, 2 × 10^7^, and 4 × 10^7^ spores/g, respectively. ^5^ The SSF conditions are as follows: moisture content = 80%, spore concentration = 4 × 10^7^ spores/g, pH = 4.0, incubation temperature = 30, and 35 °C, fermentation time = 60 h, 96, 144, and 192 respectively.

**Table 2 molecules-27-08949-t002:** Effects of solid-state fermentation conditions on the phenolic contents (mg/g) of *Citrus reticulata* peels.

Samples	Variable	Ferulic Acid	Narirutin	Hesperidin	Nobiletin	Tangeretin
pH ^1^						
Control	-	0.46 ± 0.00 ^a^	4.97 ± 0.07 ^b^	19.36 ± 0.47 ^a^	6.31 ± 0.11 ^a^	2.91 ± 0.04 ^a^
FC1a	4.0	0.45 ± 0.01 ^a^	5.53 ± 0.13 ^a^	19.69 ± 0.13 ^a^	6.36 ± 0.08 ^a^	2.91 ± 0.02 ^a^
FC1b	4.5	ND	4.25 ± 0.22 ^c^	13.83 ± 0.04 ^b^	5.63 ± 0.20 ^b^	2.59 ± 0.09 ^b^
FC1c	5.0	ND	4.23 ± 0.03 ^c^	9.28 ± 1.13 ^c^	5.76 ± 0.05 ^b^	2.64 ± 0.02 ^b^
FC1d	5.5	ND	4.25 ± 0.04 ^c^	7.46 ± 0.24 ^c^	5.73 ± 0.03 ^b^	2.67 ± 0.01 ^b^
FC1e	6.0	0.31 ± 0.00 ^b^	4.30 ± 0.16 ^c^	15.39 ± 0.72 ^b^	5.77 ± 0.05 ^b^	2.66 ± 0.02 ^b^
FC1f	6.5	0.24 ± 0.01 ^c^	4.17 ± 0.09 ^c^	8.41 ± 0.96 ^c^	5.56 ± 0.31 ^b^	2.54 ± 0.14 ^b^
Incubation temperature (IT, °C) ^2^						
Control	-	0.46 ± 0.00 ^a^	4.97 ± 0.07 ^bc^	19.36 ± 0.47 ^a^	6.31 ± 0.11 ^a^	2.91 ± 0.04 ^a^
FC2a	25	0.43 ± 0.02 ^a^	4.48 ± 0.33 ^c^	15.02 ± 0.36 ^b^	5.71 ± 0.05 ^b^	2.60 ± 0.03 ^b^
FC2b	30	0.45 ± 0.01 ^a^	5.53 ± 0.13 ^ab^	19.69 ± 0.13 ^a^	6.36 ± 0.08 ^a^	2.91 ± 0.02 ^a^
FC2c	35	0.41 ± 0.03 ^a^	5.76 ± 0.30 ^a^	16.41 ± 1.10 ^b^	5.45 ± 0.01 ^c^	2.52 ± 0.02 ^c^
Moisture content (MC,%, w.b.) ^3^						
Control	-	0.46 ± 0.00 ^b^	4.97 ± 0.07 ^b^	19.36 ± 0.47 ^c^	6.31 ± 0.11 ^a^	2.91 ± 0.04 ^a^
FC3a	70	0.45 ± 0.01 ^b^	5.53 ± 0.13 ^a^	19.69 ± 0.13 ^bc^	6.36 ± 0.08 ^a^	2.91 ± 0.02 ^a^
FC3b	80	0.53 ± 0.01 ^a^	5.41 ± 0.25 ^a^	22.73 ± 0.38 ^a^	6.27 ± 0.05 ^a^	2.63 ± 0.03 ^b^
FC3c	90	0.39 ± 0.01 ^c^	4.30 ± 0.14 ^c^	20.42 ± 0.47 ^b^	5.91 ± 0.15 ^b^	2.57 ± 0.07 ^b^
Spore concentration (SC, spores/g) ^4^						
Control	-	0.46 ± 0.00 ^b^	4.97 ± 0.07 ^c^	19.36 ± 0.47 ^c^	6.31 ± 0.11 ^ab^	2.91 ± 0.04 ^a^
FC4a	4x10^6^	0.53 ± 0.01 ^a^	5.41 ± 0.25 ^ab^	22.73 ± 0.38 ^b^	6.27 ± 0.05 ^ab^	2.63 ± 0.03 ^b^
FC4b	2x10^7^	0.39 ± 0.0 ^c^	5.69 ± 0.05 ^a^	23.85 ± 0.69 ^ab^	6.42 ± 0.04 ^ab^	3.01 ± 0.05 ^a^
FC4c	4x10^7^	0.40 ± 0.01 ^c^	5.14 ± 0.15 ^bc^	24.89 ± 1.10 ^a^	6.46 ± 0.02 ^a^	3.05 ± 0.10 ^a^
Fermentation time (FT, h) ^5^						
Control	-	0.46 ± 0.00 ^a^	4.97 ± 0.07 ^c^	19.36 ± 0.47 ^c^	6.31 ± 0.11 ^b^	2.91 ± 0.04 ^c^
FC5a	60	0.40 ± 0.01 ^b^	5.14 ± 0.15 ^c^	24.89 ± 1.10 ^b^	6.46 ± 0.02 ^b^	3.05 ± 0.10 ^c^
FC5b	96	0.33 ± 0.00^d^	5.56 ± 0.04 ^b^	27.52 ± 0.55 ^b^	7.91 ± 0.10 ^a^	3.68 ± 0.05 ^a^
FC5c	144	0.46 ± 0.00 ^a^	6.63 ± 0.03 ^a^	27.56 ± 0.77 ^b^	7.83 ± 0.04 ^a^	3.55 ± 0.02 ^ab^
FC5d	192	0.37 ± 0.01 ^c^	4.69 ± 0.09 ^d^	28.23 ± 0.76 ^a^	7.78 ± 0.07 ^a^	3.49 ± 0.06 ^b^

Note: ND: Not detected. Means that do not share a letter are significantly different, data are expressed as mean ± SD (*n* = 3) on a dry weight basis. ^1^ The SSF conditions are as follows: incubation temperature = 30 °C, moisture content = 70%, spore concentration = 4 × 10^6^ spores/g, fermentation time = 60 h, pH = 4.0, 4.5, 5.0, 5.5, and 6.0, respectively. ^2^ The SSF conditions are as follows: moisture content = 70%, spore concentration = 4 × 10^6^ spores/g, fermentation time = 60 h, pH = 4.0, incubation temperature = 25, 30, and 35 °C, respectively. ^3^ The SSF conditions are as follows: spore concentration = 4 × 10^6^ spores/g, fermentation time = 60 h, pH = 4.0, incubation temperature = 30 °C, moisture content = 70, 80, and 90%, respectively. ^4^ The SSF conditions are as follows: moisture content = 80%, fermentation time = 60 h, pH = 4.0, incubation temperature = 30 °C, spore concentration = 4 × 10^6^, 2 × 10^7^, and 4 × 10^7^ spores/g, respectively. ^5^ The SSF conditions are as follows: moisture content = 80%, spore concentration = 4 × 10^7^ spores/g, pH = 4.0, incubation temperature = 30, and 35 °C, fermentation time = 60 h, 96, 144, and 192, respectively.

## Data Availability

The datasets generated during and/or analyzed during the current study are available from the corresponding author on reasonable request.

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
