# Peer review of "Valorization of Citrus Reticulata Peels for Flavonoids and Antioxidant Enhancement by Solid-State Fermentation Using Aspergillus niger CGMCC 3.6189"

_molecules, 2022, doi:10.3390/molecules27248949_

Round 1
Reviewer 1 Report
The scientific article entitled: “Valorization of Citrus reticulata peels for flavonoids and anti-2 oxidant enhancement by solid-state fermentation using Asper-3 gillus niger CGMCC 3.6189” is based on the study of the evaluation the ability of citrus reticulate peel powder to accumulate bioactive compounds such as phenols and flavonoids using solid-state fermentation under specific conditions of pH, fermentation temperature, % of moisture, inoculum concentration and fermentation time.
In my opinion, the article is very well structured and the design of the trial is very well thought out. The abstract gathers the main results and the introduction is very complete and well referenced. The results are clear and concise and the discussion is well argued and based on a good bibliographic support. However, as stated in the conclusions, the fermentation times need to be further refined.
Author Response
Thank you for the positive comments and useful suggestions. We will consider modifying fermentation time in future studies.
Reviewer 2 Report
The paper deals with the use of Citrus reticulata peels for flavonoids and promotion of the anti-oxidant features, by solid-state fermentation, employing Aspergillus niger CGMCC 3.6189. The paper approaches an interesting topic, and the methodologies and results are minutiously treated. Illustrative table and graphs illustrate the results. Recommended for publication, with some minor changes regarding presentation.
The Introduction deals with the characterization of Citrus genus, enhancing on Citrus reticulate species: Pericarpium citri reticulatae. The chemical compounds responsible for the anti-inflammatory, anti-diabetic, anticancer, anti-microbial, anti-viral, antioxidative, antimutagenic, anti-glycemic properties are mentioned: phenolics (flavones, flavonols, flavanones, polymethoxylflavones, and anthocyanins), carotenoids, vitamins, and fiber. Microorganisms have been used to speed up the biotransformation of flavonoids, among which A. niger is one of the most often employed.
A. niger isolated from peel of Chenpi led to the increase of the total flavonoid content and the flavonoid aglycones such as hesperetin and naringenin. Solid-state fermentation was reported as an environmentally friendly, unexpensive and easy-to-apply approach used to promote the concentration of bioactive compounds and antioxidant activity in agro-industrial wastes and plant by-products fermentation.
Methodologies are well detailed: samples preparation, preparation of the growth curve and the inoculum of Aspergillus niger CGMCC 3.6189, solid-state fermentation, analysis of total phenolic content (Folin-Ciocalteu), total flavonoid content, DPPH scavenging, HPLC for composition in phenolics, color parameters and statistical analysis (one-way ANOVA with Tukey’s test, Pearson correlation analysis and principal component analysis).
The Results section presents minutiously the effect of pH on TPC, TFC, antioxidant activity and phytochemical compositions in Citrus reticulata peel powder. Hesperidin is the most abundant flavonoid, followed by nobiletin. The effect of temperature on the chemical composition has been also investigated. In summary, the maximum flavonoid accumulation conditions were pH 4.0, temperature 30 °C, moisture content 80%, and spore concentration 4×107 spores/g d.w. for 192 h. Among the factors under study, fermentation time, spore amount, and moisture level are the most significant, impacting flavonoid build-up, antioxidant potential and the color, whereas the fermentation temperature exerted the least impact. Though long-time fermentation markedly increased the flavonoid contents and antioxidant activity, it led to a much darker color of the fruit.
The Discussion section presents comparison with the results obtained in other studies, and provides detailed justification for the results obtained. The increases in TPC, TFC, and antioxidant activity in the fermented Citrus reticulata peel powder could be assigned to the enzymes involved in the biosynthesis of flavonoids and to hydrolases.
Several presentation/language amendments:
For the phrase:
The inoculation of A. niger isolated from Chenpi to citrus peel using solid-state fermentation (SSF) increased the total flavonoid content (TFC) and the flavonoid aglycones such as hesperetin and naringenin, while the corresponding glycones hesperidin and narirutin were decreased in a much shorter period compared with the natural aging process
Suggested reformulation:
The inoculation of A. niger isolated from Citrus reticulata peel (Chenpi) using solid-state fermentation (SSF) increased the total flavonoid content (TFC) and the flavonoid aglycones such as hesperetin and naringenin, while the corresponding flavanone glycosides (hesperidin and narirutin) were decreased in a much shorter period compared with the natural aging process.
The term glycones refers to the sugar part. If the authors refer to hesperidin and narirutin, it should be marked that they represent the whole molecule (flavanone glycosides, with flavinic and sugar part).
Cyclization of chacones (cyclization of chalcones)
In Figure 2 A, assign the color of the curves and the parameters followed: L*, a*, and b*, CCI and ∆E*
Use superscript when denoting the number of spores, in all places in the Manuscript: 4x107
Very important-please check the section References, and also, ref. number in the article. Ref. 47 is in the text, but not in the final Reference list. Ref. 43 is attributed to Luo in the text, but to Chen in the final list. Ref. 44 in the text is Chen, but in the list is Shraim. Shraim et al. are at number [45] in the main body of the text.
Pay attention to the Instruction of Authors when preparing the final version.
1. Author 1, A.B.; Author 2, C.D. Title of the article. Abbreviated Journal Name Year, Volume, page range.
For instance Wang, F. ; Chen, L.
Please, use comma to separate ideas, as it is important in some cases, such as:
Liu et al. [23] reported that as the inoculum concentration increased, the flavonoid content increased first and then decreased, reaching the maximum at the inoculum concentration of....
Also, a suggested reformulation:
It is known that the increase of the inoculum concentration can shorten thefermentation time and limit the growth of other microorganisms [29].
or
It is known that increasing the inoculum concentration, can shorten the fermentation time and limit the growth of other microorganisms [29].
TE (Trolox equivalents), more rigorous than equivalent
